# Grading Nursing Care Study in Integrated Medical and Nursing Care Institution Based on Two-Stage Gray Synthetic Clustering Model under Social Network Context

**DOI:** 10.3390/ijerph191710863

**Published:** 2022-08-31

**Authors:** Lan Xu, Yu Zhang

**Affiliations:** School of Economics and Management, Jiangsu University of Science and Technology, Zhenjiang 212100, China

**Keywords:** integrated medical and nursing care services, grading nursing care, social network, two-stage gray synthetic clustering model

## Abstract

Establishing a scientific and sustainable grading nursing care evaluation system is the key to realizing the rational distribution of medical and nursing resources in the combined medical and nursing care services. This study establishes a grading nursing care index system for medical and nursing institutions from both medical and nursing aspects, and proposes a grading nursing care evaluation model based on a combination of interval-valued intuitionistic fuzzy entropy and a two- stage gray synthetic clustering model for interval gray number under a social network context. Through case analysis, the proposed method can directly classify the elderly into corresponding grading nursing care grades and realize the precise allocation of medical and nursing resources, which proves the feasibility of the method.

## 1. Introduction

Population aging has become an irreversible worldwide trend, which is often accompanied by the characteristics of chronic disease and disability. At the same time, with the accelerating pace of life, the phenomenon of empty nesting is increasing, the traditional family pension is weakening, and the problem of “bed pressure” for the elderly in hospitals is prominent. At present, the mode of separation of medical and nursing care can no longer meet the needs of the elderly [1]. “Integrated medical and nursing care services” have made an indelible contribution to the exploration of the rational allocation of medical and nursing care resources, the response to the impact of the population aging, and the integration of medical and nursing care services [2]. Many countries have explored the integrated medical and nursing care services, and the United States has provided high-quality, cost-effective community care, rehabilitation, and recreational activities for the elderly at home and the rhythm center of primary health care through PACE [3]. The UK combined the Adult Social Care System (ASC) with the National Health Service (NHS) to provide high-quality care in the areas of support, home health, and health and society for the elderly [4] (Marilyn, 2014). Japan encouraged regional and community participation in seeking healthcare through four elements: self-help (Ji-jo), mutual assistance (Go-jo), social solidarity care (Kyo-jo), and government care (Ko-jo) [5].

The integrated medical and nursing care service mode is different from the traditional home-based care, community-based care, and institutional care services. It is not a separate service mode, but introduces medical services into traditional nursing care services, thus realizing the system integration and supply of life care, disease prevention, and rehabilitation care [6]. At present, the institutions providing medical and nursing care services are mainly divided into the following categories: firstly, adding medical service content within nursing institutions; secondly, implanting nursing care services within medical institutions; thirdly, integrating medical and nursing care services deeply into the community [7,8,9].

However, there is no uniform standard for grading nursing care in numerous medical and nursing institutions in China. Most institutions are set up according to the elderly’s personal health status, economic level, and the requirements of family members. They are faced with the problems such as imperfect evaluation indices or inconsistent evaluation standards. The actual care content does not match well with the reality of the elderly. Differences between integrated medical and nursing care institutions and ordinary nursing care institutions lie in that the health condition of most elderly people is relatively poor; hence, it is significant to pay attention to rational allocation of medical resources. The grading nursing care system plays an important role in ensuring the quality of services, as well as in the rational allocation of human resources for care and the setting of fees for care services.

Grading evaluation has been applied in many fields. Xian and Guo established that the interval probability hesitant fuzzy linguistic TOPSIS model can effectively and objectively help businesses classify the strategic cooperation supplier [10]. Metin [11] classified marble blocks cut into slabs or strips using multicriteria decision methods, TOPSIS, and GRA. Hu [12] used the ADL scale to classify the elderly as having four levels of ability: intact, severe, moderate, and mildly impaired in daily living. Lu [13] constructed a grade classification model tandem BpNN method with a multi-metal sensor for rice eating quality evaluation.

For the nursing care institutions, there are also many studies that deeply discussed the grading evaluation. Li et al. [14] established a grading indicator system for long-term care of the elderly in medical and nursing care institutions in four aspects: basic action ability, physical function, mental and psychological state, and social function. Li [15] used cluster analysis and the AHP method to draw the conclusion that it is most reasonable to divide the elderly in nursing care institutions into four care levels. Shi [16] established a synthetic classification evaluation model of long-term care for the elderly from seven aspects, such as daily living ability, cognitive ability, medical care, and pressure ulcer risk. Some local governments have also established a grading nursing care evaluation system for nursing care institutions. In 2018, Shanghai put forward the “Shanghai Unified Needs Assessment and Service Management Measures for Elderly Care”, which divided the elderly into six nursing grades on the basis of the dimensions of self-care ability and disease severity. Researchers have conducted in-depth discussions on various aspects of grading nursing evaluation, but there are still some shortcomings. Firstly, studies evaluated medical and elderly services in a synthetic way, which is not conducive to the separate allocation of resources for medical and nursing care services, and which can lead to unreasonable resource allocation. Secondly, most of the studies focused on the construction of indicator systems, ignoring the importance of indicator weights for the grading nursing care. Lastly, due to the complexity and uncertainty of the elderly’s physical condition, the problem of poor information and uncertainty is widespread; hence, there is a great limitation in taking the exact figures to evaluate.

The gray clustering model can effectively solve the problem of a small sample size and poor information. Karakoç [17] used gray cluster analysis to evaluate the development level of provinces; Wu [18] proposed a multilevel gray clustering decision method for the synthetic evaluation of operation schemes of natural gas pipelines with inexact data and insufficient information. Regina [19] proposed a new methodology based on gray clustering which classifies monitoring sites according to their need for nitrogen pollution management when only small amounts of data are available. To express the evaluation objects more accurately, Ye [20] extended the application scope of gray clustering model to the category of interval gray numbers; however, when the decision objects are classified only according to the maximum value of the gray clustering coefficient vector, it is difficult to obtain reliable decision conclusions when the difference between the maximum component and other components is not significant.

To solve the above problems, Liu et al. [21] constructed a new two-stage gray synthetic clustering model, which solves the synthetic decision-making problem when the values of each component of the gray clustering coefficient vector tend to be equal or several principal components in the front row are close.

In summary, this study proposes a grading nursing care evaluation model based on a combination of interval-valued intuitionistic fuzzy entropy and interval gray number type two-stage gray synthetic clustering model under a social network context. It provides a reference for medical and nursing care institutions to improve their service efficiency and rationalize the allocation of medical and nursing care resources.

The are several contributions of this paper. Firstly, in order to realize the rational allocation of medical resources and elderly resources, a grading nursing care indicator system for the elderly is constructed from two aspects of medical treatment and nursing. Secondly, considering the social relations among decision-making experts, a large-scale group decision-making method for grading nursing care indicators of medical and nursing care institutions based on interval-valued intuitionistic fuzzy entropy under a social network context is constructed. Thirdly, in the two-stage gray synthetic clustering model, interval gray numbers are introduced to describe the complexity and uncertainty of the elderly’s physical condition.

## 2. Methodology

### 2.1. Determination of Grading Nursing Care Indicators Weight Based on Hesitation and Ambiguity under Social Network Context

The determination of weight will have a significant influence on grading nursing care evaluation; therefore, many experts in various fields need to judge the importance of indicators in accordance with their own knowledge. To deal with the complex social relations among decision makers, the concept of social network has been gradually applied to group decision making [22,23].

#### 2.1.1. Community Network Division Based on Fast Unfolding Algorithm

Community detection algorithms are a powerful tool for partitioning social networks. The Fast-Unfolding algorithm, a cohesive clustering algorithm based on modularity optimality proposed by Blondel [24], can be used to solve large-scale social network classification problems.

Given a network with multiple nodes, the Fast-Unfolding algorithm proceeds as follows: (1) every node is regarded as an independent community; (2) for each node $p_{i}$, it is divided into adjacent communities in turn, the modularity gain ΔQ after subdivision is calculated, and then the node $p_{i}$ is placed in the community with the greatest benefit. If ΔQ < 0, the node $p_{i}$ remains in the original community; (3) the steps are repeated until the community of the network no longer changes; (4) the nodes are merged in the community into one node to build a hierarchical network, and the above steps are performed until the modularity no longer increases. The modularity gain ΔQ is expressed as
(1)$$\Delta Q=\left[\frac{\sum in+2p_{i,in}}{2m}-{\left(\frac{\sum tot+p_{i}}{2m}\right)}^{2}\right]-\left[\frac{\sum in}{2m}-{\left(\frac{\sum tot}{2m}\right)}^{2}-{\left(\frac{p_{i}}{2m}\right)}^{2}\right],$$
where ∑in represents the total degree of all edges of the community, $p_{i,in}$ represents the total number of edges from node $p_{i}$ to all nodes in the community, m represents the total number of edges in the community, and ∑tot represents the weight sum of edges connected with node $p_{i}$ in the community.

#### 2.1.2. Determination of the Weight of Social Community Nodes

The weight determination of social nodes is mainly measured by the structural location attribute of nodes in the whole network [25]. Degree centrality is the most direct indicator of the degree centrality of nodes. More connected nodes are more important. Closeness centrality reflects how close the node is to other nodes. Closer nodes indicate a greater impact. The PageRank value of a node not only considers the influence of adjacent nodes on the importance of the node, but also considers the promoting effect of nonadjacent nodes on the importance of the node. Therefore, this study determines the weight of nodes by combining the centrality and PR value of nodes, and the specific steps are described below.

Step 1: Compute the degree centrality of nodes.
(2)$$C_{d}\left(p_{i}\right)=\underset{j}{\sum }N_{ij}.$$

The degree centrality $C_{d}\left(p_{i}\right)$ of node $p_{i}$ is the number of adjacent nodes of $p_{i}$. $N_{ij}$ indicates that there is an edge connection between node $p_{i}$ and node $p_{j}$, whereby $N_{ij}$ = 1; otherwise, it is 0.

Step 2: Compute the closeness centrality of nodes.
(3)$$C_{c}\left(p_{i}\right)=\frac{1}{\sum_{j}MinDist\left(p_{i},p_{j}\right)}.$$

The closeness centrality $C_{c}\left(p_{i}\right)$ of the node $p_{i}$ represents the reciprocal of the sum of the shortest paths from $p_{i}$ to other nodes in the community.

Step 3: Calculate the PageRank value of nodes.
(4)$$PR_{p_{i}}=d*\underset{p_{j}}{\sum }\frac{PR\left(p_{j}\right)}{N\left(p_{s}\right)}+\left(1-d\right),$$
where d is damping coefficient, and its value range is [0,1]. $N\left(p_{s}\right)$ is the number connected with the node $p_{i}$. $PR\left(p_{i}\right)$ indicates the PR value of node $p_{i}$.

Step 4: Calculate the importance of nodes $C_{mix}\left(p_{i}\right)$.
(5)$$C_{mix}\left(p_{i}\right)=\alpha \frac{C_{d}\left(p_{i}\right)}{\sum_{t=1}^{n}C_{d}\left(p_{i}\right)}+\beta \frac{C_{c}\left(p_{i}\right)}{\sum_{t=1}^{n}C_{c}\left(p_{i}\right)}+\gamma \frac{PR_{p_{i}}}{\sum_{t=1}^{n}PR_{p_{i}}},$$
where α, β, and γ respectively represent the adjustment coefficient of degree centrality, closeness centrality, and the PR value, α, β, γ∈[0, 1], and α+β+γ=1.

Step 5: Calculate the weight of nodes.

Using the Fast-Unfolding algorithm, the M nodes in the social network are divided into K communities, and the partition structure is $\left\{C^{k}丨k=1,2,\cdots ,K\right\}$. The weight $w_{p}$ of node $p_{i}$ is expressed as
(6)$$w_{p}=\frac{C_{mix}\left(p_{i}\right)}{\sum_{t=1}^{n_{k}}C_{mix}\left(p_{i}\right)},$$
where $n_{k}$ represents the number of nodes in k partition.

#### 2.1.3. Determination of Partition Weight Based on Social Network

The weight of a partition is measured by the distance between the mixed center of the partition and the mixed center of the whole network (the greater the distance, the smaller the weight), which is calculated using the method described in [26].

Step 6: Calculate the normalized weight $\lambda_{k}$ of partition.
(7)$$\lambda_{k}=\frac{1}{\left|C_{H}^{k}-C_{H}\right|},$$
where $C_{H}^{k}$ is the synthetic importance of the k-th partition, which is obtained from the average value of the importance of all nodes in the partition. $C_{H}$ represents the synthetic importance of the whole network, which is equal to the average value of the importance of all nodes in the network.

Step 7: Normalize the weight $w_{k}$ of partition $C^{k}$.
(8)$$w_{k}=\frac{\lambda_{k}}{\sum_{k=1}^{K}\lambda_{k}}.$$

#### 2.1.4. Determination of Indicator Weights under Ambiguity and Hesitation Context

Given that most of the grading nursing care indicators of medical and nursing care institutions are qualitative indicators, it is difficult to express decision information with precise numbers due to the complexity of environment and uncertainty of decision makers’ cognition. Therefore, this study introduces interval-valued intuitionistic fuzzy number into the determination of grading nursing care indicator weights, and then describes the fuzziness and uncertainty of information from four aspects: membership, non-membership, ambiguity, and hesitation, thus reducing the loss of decision information.

Let A be the IVIFS in the universe of discourse X as follows:(9)$$A=\left\{〈x,\;\left[\mu_{x}^{L},\mu_{x}^{U}\right],\left[\nu_{x}^{L},\nu_{x}^{U}\right]〉丨x\in X\right\},$$
where membership degree $\mu_{x}$ = [$\mu_{x}^{L},\mu_{x}^{U}$]∈[0,1], non-membership degree $\nu_{x}$ = [$\nu_{x}^{L},\nu_{x}^{U}$] ∈[0,1], and $\mu_{x}^{L}<\mu_{x}^{U}$, $\nu_{x}^{L}<\nu_{x}^{U}$, $\mu_{x}^{U}+\nu_{x}^{U}\leq 1$.

Step 8: Set the decision matrix as follows:(10)$$[\begin{array}{cccc} {}[(a_{1},b_{1}),(c_{1},d_{1})]w_{1} & [(a_{2},b_{2}),(c_{2},d_{2})]w_{1} & \cdots & [(a_{k},b_{k}),(c_{k},d_{k})]w_{1}\\ {}[(a_{k},b_{k}),(c_{k},d_{k})]w_{2} & [(a_{2},b_{2}),(c_{2},d_{2})]w_{2} & \cdots & [(a_{k},b_{k}),(c_{k},d_{k})]w_{2}\\ \vdots & \vdots & \ddots & \vdots \\ {}[(a_{1},b_{1}),(c_{1},d_{1})]w_{s} & [(a_{2},b_{2}),(c_{2},d_{2})]w_{s} & \cdots & [(a_{k},b_{k}),(c_{k},d_{k})]w_{s} \end{array}].$$

Step 9: The ambiguity of IVIFE is defined by the distance relationship between interval numbers; thus, the ambiguity $\hat{\Lambda }\left(x\right)$ of x is expressed as
(11)$$\hat{\Lambda }\left(x\right)=\frac{\sqrt{2}}{2}dist\left(\hat{\mu }\left(x\right),\hat{\nu }\left(x\right)\right)=\frac{\sqrt{2}}{2}\sqrt{{\left({\hat{\mu }}^{L}\left(x\right)-{\hat{\nu }}^{L}\left(x\right)\right)}^{2}+{\left({\hat{\mu }}^{U}\left(x\right)-{\hat{\nu }}^{U}\left(x\right)\right)}^{2}},$$
where $dist\left(\hat{\mu }\left(x\right),\hat{\nu }\left(x\right)\right)$ represents the Euclidean distance between interval numbers [27] (Ji 2021), which can better reflect the ambiguity.

Step 10: Determine the hesitation degree.
(12)$$\begin{array}{ll} \hat{\pi }\left(x\right)=1-\hat{\mu }\left(x\right) & -\hat{\nu }\left(x\right)=\left[{\hat{\pi }}^{L}\left(x\right),{\hat{\pi }}^{U}\left(x\right)\right]\\ & =\left[\left(1-\mu^{U}\left(x\right)-\nu^{U}\left(x\right)\right),\left(1-\mu^{L}\left(x\right)-\nu^{L}\left(x\right)\right)\right], \end{array}$$
where $\hat{\pi }\left(x\right)$ represents the hesitation degree in IVIFS; if $\mu^{U}\left(x\right)$ = $\nu^{U}\left(x\right)$, $\mu^{L}\left(x\right)$ = $\nu^{L}\left(x\right)$, the hesitation degree is 0, and the IVIFS turns into intuitionistic fuzzy set (IFS).

Step 11: According to the entropy formula of IVIFS developed by Guo [28], the ambiguity degree is improved by Equation (11), and the equation of interval-values intuitionistic fuzzy entropy E(x) is expressed as
(13)$$E\left(x\right)=\frac{1}{n}\sum_{i=1}^{n}\left[1-\frac{\sqrt{2}}{2}\sqrt{{\left({\hat{\mu }}^{L}\left(x\right)-{\hat{\nu }}^{L}\left(x\right)\right)}^{2}+{\left({\hat{\mu }}^{U}\left(x\right)-{\hat{\nu }}^{U}\left(x\right)\right)}^{2}}\right]\frac{1+\frac{1}{2}\left({\hat{\pi }}^{L}\left(x\right)+{\hat{\pi }}^{U}\left(x\right)\right)}{2}.$$

Step 12: Determine the indicator weights $\omega_{j}$.
(14)$$\omega_{j}=\frac{1-E\left(x\right)}{s-\sum_{j=1}^{s}E\left(x\right)}.$$

### 2.2. Two-Stage Gray Synthetic Clustering Grading Evaluation Model for Interval Gray Number Based on Kernel and Degree of Grayness

In the process of obtaining data, when judging the physical condition of the elderly, due to the subjectivity of medical staff and the instability of the elderly’s health condition, the observed data have strong complexity and uncertainty, and it is often impossible to get specific values, which can be considered “gray”. For this reason, this paper adopts interval gray number assignment, which is closer to reality. Through scoring by experts, the interval gray value of each indicator of the elderly and the whitenization weight function value were obtained. The universe of the indicator value and whitenization weight function value is [0, 100]. According to the interval gray number operation theorem of kernel and degree of grayness, the gray number operation is converted into real number operation, and the problem of interval gray number operation is solved to some extent [29]. The proposed method is concise, with low computational overhead, and it fully considers the characteristics of interval gray number, which makes the decision result more reasonable.

#### 2.2.1. The First Stage of the Two-Stage Gray Synthetic Clustering Model

Step 1: Calculate whitenization weight function. Divide into s gray classes according to evaluation requirements and determine the turning point of gray class 1 to gray class s as ${⨂}_{j}^{k}\left(1\right)$ to ${⨂}_{j}^{k}\left(s\right)$, where ${⨂}_{j}^{k}\left(s\right)\in \left[{⨂}_{j}^{k}{\left(s\right)}^{-},{⨂}_{j}^{k}{\left(s\right)}^{+}\right]$, and ${\hat{⨂}}_{ij}$ represents the kernel of the gray number. For any ${\hat{⨂}}_{ij}$, there is a whitenization weight function $f\left({⨂}_{ij}\right)$ relative to the gray class k, which can be described as
(15)$$f\left({⨂}_{ij}\right)=\{\begin{array}{c} {\left(\frac{{\hat{⨂}}_{ij}-\hat{⨂}\left(k+1\right)}{\hat{⨂}\left(k\right)-\hat{⨂}\left(k+1\right)}\right)}_{\left(g_{ij}^{ο}⋁g^{ο}\left(k\right)⋁g^{ο}\left(k+1\right)\right)},{\hat{⨂}}_{ij}\in \left(\hat{⨂}\left(k\right),\hat{⨂}\left(k+1\right)\right)\\ {\left(\frac{\hat{⨂}\left(k-1\right)-{\hat{⨂}}_{ij}}{\hat{⨂}\left(k-1\right)-\hat{⨂}\left(k\right)}\right)}_{\left(g_{ij}^{ο}⋁g^{ο}\left(k-1\right)⋁g^{ο}\left(k\right)\right)},\;{\hat{⨂}}_{ij}\in \left(\hat{⨂}\left(k-1\right),\hat{⨂}\left(k\right)\right) \end{array},$$
where ${\hat{⨂}}_{ij}$ represents the kernel for the gray number.
(16)$${\hat{⨂}}_{ij}=\frac{1}{2}\left({⨂}_{ij}^{-}+{⨂}_{ij}^{+}\right),$$
where ${⨂}_{ij}\in \left[{⨂}_{ij}{}^{-},{⨂}_{ij}{}^{+}\right]$ is the observed value of index j corresponding to object i, and $\omega_{j}$ is the weight of index j.

Step 2: Calculate the gray comprehensive clustering coefficient for interval gray number $\sigma_{i}^{k}$.
(17)$$\sigma_{i}^{k}={\left[\sum_{j=1}^{m}f\left({⨂}_{ij}\right)\cdot \omega_{j}\right]}_{g_{ij}^{ο}},$$
where $g_{ij}^{ο}$ represents the degree of grayness for gray number ${⨂}_{ij}$.
(18)$$g_{ij}^{ο}=\frac{\mu \left({⨂}_{ij}\right)}{\mu \left(\Omega_{ij}\right)},$$
where the background or universe of gray number ${⨂}_{ij}$ is Ω, and $\mu \left({⨂}_{ij}\right)$ is the measure of the universe of gray number ${⨂}_{ij}$.

Step 3: Determine the gray classes of the object. From {σik}1≤k≤smax=σik*, it is judged that the object i belongs to the gray class $k^{*}$.

#### 2.2.2. The Second Stage of the Two-Stage Gray Synthetic Clustering Model

When the maximum component of the gray clustering coefficient vector differs greatly from the other components, the decision results are reliable [30]. However, when the difference between the maximum component and the other components is small, the decision results are not necessarily reliable. Consequently, we need to consider the gray clustering coefficient vector as a whole for synthetic evaluation.

Step 4: Calculate the gray synthetic weighted decision-making vector $\eta_{k}$.
(19)$$\eta_{k}=\left\{\frac{1}{\frac{s\left(s+1\right)}{s}+\left[\left(k-1\right)s-\frac{k\left(k-1\right)}{2}\right]}\right\}\cdot \left(s-k+1,s-k+1,\cdots ,s-1,s,s-1,\cdots ,k\right).$$

Step 5: Calculate the weighted decision-making coefficient vector of kernel clustering $\chi_{\sigma_{i}^{k}}$.
(20)$$\chi_{\sigma_{i}^{k}}=\eta_{k}\cdot \sigma_{i}^{k}.$$
when $\chi_{\sigma_{1}^{k}=}\chi_{\sigma_{2}^{k}}$, if $g_{\sigma 1}^{k}=g_{\sigma 2}^{k}$, then $\sigma_{1}^{k}=\sigma_{2}^{k}$; if $g_{\sigma 1}^{k}>g_{\sigma 2}^{k}$, then $\sigma_{1}^{k}<\sigma_{2}^{k}$; if $g_{\sigma 1}^{k}<g_{\sigma 2}^{k}$, then $\sigma_{1}^{k}>\sigma_{2}^{k}$.

## 3. Case Study

### 3.1. Construction and Weight Determination of Grading Nursing Care Indicator System

At present, there is no unified standard and basis for the classification of nursing grades in pension institutions in China. Beijing, Shanghai, and other cities have formulated and released the local evaluation standards for the integrated medical and nursing care services, but the research on its graded nursing is relatively backward. According to local standards, combined with the literature [31,32,33] and the characteristics of the elderly in integrated medical and nursing care institutions, this study constructed the grading nursing care evaluation indicator system in medical and nursing institutions. The social network was used to determine the weights of decision makers, and the interval-valued intuitionistic fuzzy method was used to determine the weights of decision makers. Finally, a two-stage gray synthetic clustering grading evaluation model was used to determine the care level of the elderly. The social network provides a good solution for the determination of expert weight in interval intuitionistic fuzzy method. The combination of the above methods reduces the uncertainty of the index weight in the gray model to some extent. Therefore, the combination of the above three methods has good applicability for the application scenario of this study. The specific steps were shown in Section 2.

A total of 20 experts in the field of integrated medical and nursing care services were invited to score the grading nursing care indicators. Experts and indicators are expressed by $E=\left\{E_{1},E_{2},\cdots ,E_{20}\right\}$ and $I=\left\{I_{1},I_{2},\cdots ,I_{8}\right\}$ respectively. Taking experts as nodes and their social relations as edges, through the Fast-Unfolding community detection algorithm, the partition results are shown in Figure 1.

The weight of each partition and expert was calculated using the partition results and the method proposed in Section 3.1 (see Table 1). The weights of grading nursing care evaluation indicators for integrated medical and nursing care services are shown in Table 2.

### 3.2. Grading Nursing Care Research

#### 3.2.1. The First Stage of Grading Nursing Care Evaluation

Li [15], through K-means cluster analysis, believes that the classification result is the best when the care level of the elderly in integrated medical and nursing care institutions is divided into four categories. Through the Delphi method, four types of interval gray number possibility function values were determined, which correspond to four gray classes (k=1, 2, 3, 4): medical/nursing care grade I (slight), medical/nursing grade II (medium), medical/nursing grade III (high), and medical/nursing grade IV (very high). The possibility function values of gray k are as follows:$$f_{j}^{1}\left[\left[-\right],\left[-\right],\left[34,36\right],\left[53,57\right]\right],$$
$$f_{j}^{2}\left[\left[34,36\right],\left[53,57\right],\left[-\right],\left[72,75\right]\right],$$
$$f_{j}^{3}\left[\left[53,57],[72,75\right],\left[-\right],\left[83,85\right]\right],$$
$$f_{j}^{4}\left[\left[72,75],[83,85\right],\left[-\right],\left[-\right]\right].$$

The universe of the indicator value function is [0, 100].

In this study, five residents in an integrated medical and nursing care institution were taken as examples, and the interval gray value of each indicator was obtained by scoring (the higher the score, the more serious the situation of the elderly in the indicator) of appraisers who have the experience in medical and nursing care service (medical care appraisers need to obtain the qualification of medical practitioner or practicing assistant doctor). The interval gray number matrix P(⨂) was as follows:
$$\begin{array}{lll} P\left(⨂\right) & & \\ =\{\begin{array}{cccc} \left[65,70\right] & \left[25,30\right] & \left[50,56\right] & \left[40,42\right]\\ \left[30,36\right] & \left[90,93\right] & \left[75,79\right] & \left[80,82\right]\\ \left[78,82\right] & \left[35,37\right] & \left[82,85\right] & \left[77,79\right]\\ \left[45,49\right] & \left[58,60\right] & \left[45,51\right] & \left[40,44\right]\\ \left[48,52\right] & \left[62,65\right] & \left[73,77\right] & \left[50,54\right] \end{array} & | & \begin{array}{cccc} \left[20,30\right] & \left[10,20\right] & \left[65,70\right] & \left[40,47\right]\\ \left[40,45\right] & \left[12,15\right] & \left[65,67\right] & \left[75,78\right]\\ \left[75,79\right] & \left[32,40\right] & \left[75,77\right] & \left[80,84\right]\\ \left[50,52\right] & \left[16,20\right] & \left[90,95\right] & \left[82,84\right]\\ \left[80,85\right] & \left[72,80\right] & \left[56,58\right] & \left[70,74\right] \end{array}\} \end{array}.$$


According to the interval gray values and weights for each indicator, using the constructed whitening weight function and Equation (17), the whitening weight function values and gray clustering coefficients of each indicator in different gray classes could be calculated, as shown in Table 3.

Analyzing the results in Table 3, it can be seen from {σimk}1≤k≤4max=σ1m2=0.448(0.06) that the medical care level of the first elderly was Grade II. Moreover, it can be seen from {σ1mk}1≤k≤4max=σ1c1=0.688(0.1) that the nursing care level of the first elderly was Grade Ⅰ.

#### 3.2.2. The Second Stage of Grading Nursing Care Evaluation

According to Equation (19), the synthetic weighted decision-making vector of each gray class can be obtained as follows:$$\eta_{1}=\frac{1}{10}\left(4,3,2,1\right);\;\;\eta_{2}=\frac{1}{12}\left(3,4,3,2\right);\;\;\eta_{3}=\frac{1}{12}\left(2,3,4,3\right);\;\;\eta_{4}=\frac{1}{10}\left(1,2,3,4\right).$$

Through Equation (20), we can get the weighted decision-making coefficient vector of kernel clustering of medicine and nursing care as follows:

$p_{1}Medical$:$$\chi_{1}=\left(\chi_{\sigma_{i}^{k}}^{1},\chi_{\sigma_{i}^{k}}^{2},\chi_{\sigma_{i}^{k}}^{3},\chi_{\sigma_{i}^{k}}^{4}\right)=\left(0.31_{\left(0.06\right)},0.287_{\left(0.06\right)},0.242_{\left(0.06\right)},0.19_{\left(0.06\right)}\right);$$

$p_{1}Nursing$:$$\chi_{1}^{∲}=\left(\chi_{\sigma_{i}^{k}}^{1},\chi_{\sigma_{i}^{k}}^{2},\chi_{\sigma_{i}^{k}}^{3},\chi_{\sigma_{i}^{k}}^{4}\right)=\left(0.356_{\left(0.1\right)},0.265_{\left(0.1\right)},0.204_{\left(0.1\right)},0.144_{\left(0.1\right)}\right).$$

After calculating the synthetic decision-making coefficient vector and reclassifying, the medical care level of the elderly $p_{1}$ changed from Grade Ⅱ to Ⅰ, and the nursing care level remained Grade Ⅰ. The change in medical care grade shows that it can meet the standard of medical care Grade I after integrating the various gray classes. Therefore, on the premise of ensuring the needs of the elderly for medical services, it can effectively save medical resources and promote the rational allocation of medical resources.

As above, the weighted decision-making coefficient vectors of kernel clustering of the remaining elderly are shown in Table 4.

From the maximum weighted decision-making coefficient vector in Table 4, the grading nursing care level of each the elderly can be obtained as shown in Figure 2.

As shown in Figure 2, the medical care level of $p_{2}$ was Grade IV, and the nursing care level was Grade I, which indicates that more energy needs to be devoted to medical care during the care process, thus saving some endowment resources. The medical care level of $p_{5}$ was Grade II, and the nursing care level was Grade IV, which indicates that the elderly need to spend more endowment resources.

Ordering the above results, the gray class of the object was judged according to weighted decision-making coefficient vector, and the grading nursing care level was determined. For example, both $p_{1}Medical$ and $p_{4}Medical$ belonged to the gray class I, the gray class I of $p_{4}$ was significantly greater than that of $p_{1}$, and the gray classes III and IV of $p_{4}$ were obviously lower than those of $p_{1}$. To sum up, the medical care needs of $p_{4}$ were obviously lower than $p_{1}$; thus, $p_{1}>p_{4}$ in rank.

Through the above analysis, the final ranking results were as follows:$$Medical\;aspect:p_{3}>p_{2}>p_{5}>p_{1}>p_{4};$$
$$Nursing\;care\;aspect:p_{5}>p_{3}>p_{4}>p_{2}>p_{1}.$$

### 3.3. Comparison with TOPSIS

TOPSIS, as an evaluation method of multi-objective comprehensive decision making, has also been applied to various classification evaluation fields. For example, Wu [34] constructed an efficient and accurate method for classifying rock mass quality based on MCS–TOPSIS coupled model, while Hadi [35] presented the TOPSIS method based on Gaussian interval type-2 fuzzy sets and optimization programs to solve the inventory classification problem. The classification using the TOPSIS method requires artificial classification according to the comprehensive evaluation results. To show the effectiveness of the proposed method, this article uses the TOPSIS method [36] to evaluate the medical and nursing care of the elderly. The specific steps are as follows:
Determine the ideal solution $v^{+}$ and non-ideal solution $v^{-}$ of scheme $v_{i}$.
$$v^{+}=\left\{max\left(v_{1},v_{2},\cdots ,v_{n}\right)\right\}.$$
$$v^{-}=\left\{min\left(v_{1},v_{2},\cdots ,v_{n}\right)\right\}.$$Calculate the distance $S^{+}$ between the scheme $v_{i}$ and the ideal solution $v^{+}$.
(21)$$S^{+}=\sqrt{\overset{m}{\underset{j=1}{\sum }}\left(v_{i}-v^{+}\right)}.$$Calculate the distance $S^{-}$ between the scheme $v_{i}$ and the non-ideal solution $v^{-}$.
(22)$$S^{-}=\sqrt{\overset{m}{\underset{j=1}{\sum }}\left(v_{i}-v^{-}\right)}.$$Calculate the relative closeness $S_{i}\left(S_{i}=\frac{S^{-}}{S^{+}+S^{-}}\right)$ to the ideal solution.
(23)$$S_{i}=\frac{S^{-}}{S^{+}+S^{-}}.$$

The relative closeness values for the above five residents are as follows:

Medical aspects: $S_{p_{1}}=0.397;S_{p_{2}}=0.655;S_{p_{3}}=0.712;S_{p_{4}}=0.416;S_{p_{5}}=0.622;$
$$Medical\;ranking:p_{3}>p_{2}>p_{5}>p_{4}>p_{1}.$$

Nursing care aspects: $S_{p_{1}}=0.083;S_{p_{2}}=0.471;S_{p_{3}}=0.809;S_{p_{4}}=0.640;S_{p_{5}}=0.839;$
$$Nursing\;care\;ranking:p_{5}>p_{3}>p_{4}>p_{2}>p_{1}.$$

Compared with Section 4.2, the ranking results of Medical were slightly different, and the ranking results of Nursing care results were the same, which proves the feasibility of the proposed method. It is easy to determine that the ranking results were different between the proposed method and the TOPSIS method. The TOPSIS method mainly focuses on the distances between the ideal solution and nonideal solution. However, the ranking results may change when adding or deleting some residents. The proposed method mainly uses the gray clustering method to divide the residents into different classes, which does not cause the phenomenon of changing ranking results [37].

## 4. Conclusions and Research Prospect

### 4.1. Conclusions

On the basis of the analysis of the characteristics of integrated medical and nursing care institutions, this study constructed a grading nursing care evaluation indicator system from two aspects: medical and nursing care. According to the social relations of decision-making members, a group decision-making method based on interval-valued intuitionistic fuzzy entropy under a social network environment was proposed to ensure the reliability of weight distribution. Finally, using a two-stage gray synthetic clustering model for interval gray number, the classification of nursing grading care of the elderly in integrated medical and nursing care institutions was realized. This study not only is conducive to the rational and accurate allocation of medical and nursing care resources, but also provides reference for the standardized development of integrated medical and nursing care services.

Through case analysis, the proposed method has the following advantages: (1) aiming at the gray quality of the elderly in the assessment process of grading nursing care, the proposed method can help institutions allocate medical and nursing care resources more accurately; (2) traditional methods tend to only rank the results, which makes it difficult to obtain more classification information. The proposed method can directly divide the elderly into different grades while providing the ranking results, which is helpful for quick classification; (3) the proposed method does not need a control group to reflect the advantages or disadvantages of the scheme; hence, it is more suitable for the reality that residents have great mobility in grading nursing assessment.

### 4.2. Research Prospect

This study provides a new way of thinking for the ability evaluation of the elderly in medical and nursing institutions, which has certain practical significance and theoretical value. At present, medical and nursing integration is in a high-speed development stage, and more mature contents will emerge in the future. Therefore, further research in the future will focus on the following aspects:(1)Customer needs. When obtaining customer needs, we can adopt the questionnaire method, which is subjective and difficult to obtain samples. Therefore, it is necessary to combine big data technology, regional differences, gender differences, economic differences, etc. to establish a multidimensional customer needs analysis method that conforms to the characteristics of medical and nursing services.(2)Service resource planning. At present, with the limited resources, this study assessed the elderly with different abilities, which can allow studying the planning of medical care and pension resources according to their needs.

## Figures and Tables

**Figure 1 ijerph-19-10863-f001:**
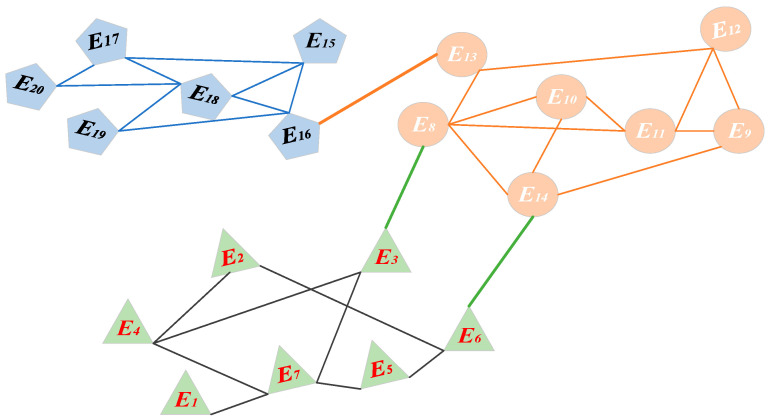
Partition result.

**Figure 2 ijerph-19-10863-f002:**
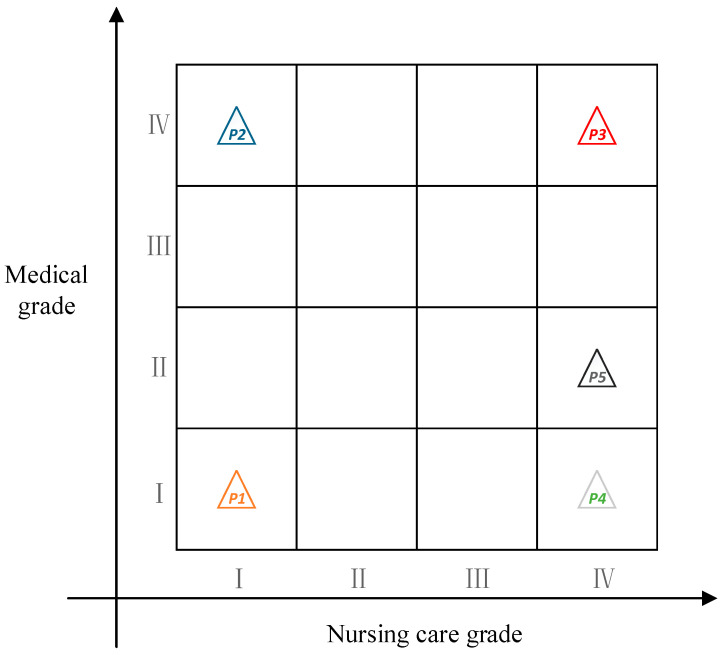
Grading nursing care level. Note:  p1–p5 represents the tested elderly, I–IV represents the care level..

**Table 1 ijerph-19-10863-t001:** The weights of partitions and experts.

Partitions	Partition Members and Their Weights							Partition Weight
${\mathrm{Community}}_{1}$	$E_{1}$	$E_{2}$	$E_{3}$	$E_{4}$	$E_{5}$	$E_{6}$	$E_{7}$	0.32
Weight	0.08	0.12	0.16	0.16	0.12	0.16	0.20	
${\mathrm{Community}}_{2}$	$E_{8}$	$E_{9}$	$E_{10}$	$E_{11}$	$E_{12}$	$E_{13}$	$E_{14}$	0.31
Weight	0.19	0.12	0.13	0.15	0.13	0.13	0.15	
${\mathrm{Community}}_{3}$	$E_{15}$	$E_{16}$	$E_{17}$	$E_{18}$	$E_{19}$	$E_{20}$		0.37
Weight	0.16	0.21	0.15	0.23	0.13	0.12		

**Table 2 ijerph-19-10863-t002:** Grading nursing care evaluation indicators and weights.

Type	Indicators		Description	Weights
Medical	$I_{1}$	Severity of acute disease	Sudden diseases or signs such as heart disease and epilepsy	0.335
	$I_{2}$	Severity of chronic disease	Chronic disease monitoring such as diabetes and hypertension.	0.241
	$I_{3}$	Rehabilitation nursing needs	Rehabilitation training, regular dressing change, infusion, etc.	0.215
	$I_{4}$	Stability of illness	The condition is stable and improving, during the rehabilitation period.	0.209
Nursing care	$I_{5}$	Basic activity abilities	Ability to eat, bathe, dress, and go to the toilet.	0.336
	$I_{6}$	Cognitive abilities	Forgetfulness, lack of judgment, dementia, etc.	0.193
	$I_{7}$	Abnormal behavior	Abnormal behaviors such as moodiness, lack of control, and outlook on life and death	0.194
	$I_{8}$	Physiological function	Audiovisual function, communication ability, self-care ability	0.277

**Table 3 ijerph-19-10863-t003:** Gray clustering coefficients.

Medical	[34, 36]	[53,57]	[72,75]	[83,85]	Nursing	[34,36]	[53,57]	[72,75]	[83,85]
1		0.324_(0.05)_	0.676_(0.05)_		1	1_(0.1)_			
2	1_(0.05)_				2	1_(0.1)_			
3	0.1_(0.06)_	0.9_(0.06)_			3		0.324_(0.05)_	0.676_(0.05)_	
4	0.3_(0.04)_	0.7_(0.04)_			4	0.575_(0.07)_	0.425_(0.07)_		
$\sigma_{1m}^{k}$	0.325_(0.06)_	0.448_(0.06)_	0.226_(0.05)_	0	$\sigma_{1c}^{k}$	0.688_(0.1)_	0.181_(0.07)_	0.131_(0.05)_	0

**Table 4 ijerph-19-10863-t004:** The weighted decision-making coefficient vectors.

	Medical	Nursing
$p_{1}$	$Max\left(0.31_{\left(0.06\right)},0.287_{\left(0.06\right)},0.242_{\left(0.06\right)},0.19_{\left(0.06\right)}\right)$	$Max\left(0.356_{\left(0.1\right)},0.265_{\left(0.1\right)},0.204_{\left(0.1\right)},0.144_{\left(0.1\right)}\right)$
$p_{2}$	$Max\left(0.221_{\left(0.06\right)},0.212_{\left(0.06\right)},0.239_{\left(0.06\right)},0.279_{\left(0.06\right)}\right)$	$Max\left(0.295_{\left(0.05\right)},0.259_{\left(0.05\right)},0.241_{\left(0.05\right)},0.205_{\left(0.05\right)}\right)$
$p_{3}$	$Max\left(0.197_{\left(0.04\right)},0.209_{\left(0.04\right)},0.252_{\left(0.04\right)},0.303_{\left(0.04\right)}\right)$	$Max\left(0.199_{\left(0.08\right)},0.219_{\left(0.08\right)},0.270_{\left(0.08\right)},0.301_{\left(0.08\right)}\right)$
$p_{4}$	$Max\left(0.329_{\left(0.06\right)},0.300_{\left(0.06\right)},0.226_{\left(0.06\right)},0.171_{\left(0.06\right)}\right)$	$Max\left(0.234_{\left(0.05\right)},0.235_{\left(0.05\right)},0.231_{\left(0.05\right)},0.266_{\left(0.05\right)}\right)$
$p_{5}$	$Max\left(0.276_{\left(0.04\right)},0.294_{\left(0.04\right)},0.264_{\left(0.04\right)},0.223_{\left(0.04\right)}\right)$	$Max\left(0.186_{\left(0.08\right)},0.238_{\left(0.08\right)},0.289_{\left(0.08\right)},0.314_{\left(0.08\right)}\right)$

## Data Availability

The data that support the findings of this study are available from the corresponding author, upon reasonable request.

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
