# Peer review of "Grading Nursing Care Study in Integrated Medical and Nursing Care Institution Based on Two-Stage Gray Synthetic Clustering Model under Social Network Context"

_ijerph, 2022, doi:10.3390/ijerph191710863_

Round 1

Reviewer 1 Report

1. The literature review section should include information on the development of integrated medical and nursing care in a few typical developed countries. Additionally, the Integrated Medical and Nursing Care Institutions' current grading system should be implemented.

2. What facets of the research background are grey according to the grey method? Why adopting the grey theory in this situation is suitable? Please describe why the grey theory is better than other methods.

3. In comparison to the general grey comprehensive measurement method, what are the benefits of the two-stage grey synthetic clustering grading evaluation model for interval grey number based on kernel and degree of greyness? Why do you employ the enhanced method?

4. Why are you interested in comparing techniques with TOPSIS? Is the comparison sufficient to show the advantage of the proposed method of this paper?

Author Response

Thank you very much for giving us an opportunity to revise our manuscript. We appreciate the editor and reviewers very much for their constructive comments and suggestions on our manuscript entitled“Grading Nursing Care Study in Integrated Medical and Nursing Care Institution Based on Two Stage Grey Synthetic Clustering Model under Social Network Context”.The main corrections in the paper and the responds to the reviewer’s comments are as flowing:

  1. The literature review section should include information on the development of integrated medical and nursing care in a few typical developed countries. Additionally, the Integrated Medical and Nursing Care Institutions' current grading system should be implemented.

Response:Thank you for your suggestion. The authors have found and added a section on the progress of medical care in developed countries in the literature review section, as follows:

Many countries have explored the integrated medical and nursing care services, and the United States has provided high-quality, cost-effective community care, rehabilitation and recreational activities for the elderly at home and the rhythm center of primary health care through PACE. The UK combined the Adult Social Care System (ASC) with the National Health Service (NHS) to provide high-quality care in the areas of support, home health, and health and society for the elderly. Japan encouraged regional and community participation in seeking health care through four elements: self-help (Ji-jo), mutual assistance (Go-jo), social solidarity care (Kyo-jo) and government care (Ko-jo) [6].

  1. What facets of the research background are grey according to the grey method? Why adopting the grey theory in this situation is suitable? Please describe why the grey theory is better than other methods.

Response:In the process of obtaining data, when judging the physical condition of the elderly, due to the subjectivity of medical staff and the instability of the elderly's health condition, the observed data show strong complexity and uncertainty, and it is often impossible to get specific values, which is called “grey”. Relative explanation has been provided in section 2.2 in revised version of the manuscript.

  1. In comparison to the general grey comprehensive measurement method, what are the benefits of the two-stage grey synthetic clustering grading evaluation model for interval grey number based on kernel and degree of greyness? Why do you employ the enhanced method?

Response:Thank you for your comments. To address your concerns,The authors have added the corresponding contents and marked them in red in Section 2.2.

Based on the interval grey number operation theorem of kernel and degree of greyness, the grey number operation is converted into real number operation, and the problem of interval grey number operation is solved to some extent(Liu, 2010).The proposed method is concise, low computational overhead, and fully considers the characteristics of interval grey number, which makes the decision result more reasonable.

  1. Why are you interested in comparing techniques with TOPSIS? Is the comparison sufficient to show the advantage of the proposed method of this paper?

Response:The authors have added literature review at the top of section 4.4. TOPSIS method is also a multi-objective decision-making method, which has been applied to many fields, such as rock mass classification and inventory classification, and has a good application effect. Therefore, the rationality of the proposed method can be shown by comparison with it.

Reviewer 2 Report

With respect to the evaluation problems of sustainable grading nursing care in order to realize the rational distribution of medical and nursing resources in the combined medical and nursing care services, this paper establishes a grading nursing care index system for medical and nursing institutions from both medical and nursing aspects, and proposes a grading nursing care evaluation model based on a combination of interval-valued intuitionistic fuzzy entropy and two- stage grey synthetic clustering model for interval grey number under social network context, which takes on certain theoretical and practical significance. However, there exists some shortcomings as follows:

(1)  What is the rationale for combining the these approaches of interval-valued intuitionistic fuzzy entropy, two-stage grey synthetic clustering model, and social network? please give some explanations in this paper.

(2)  What is the advantages, validity and scope of the model constructed? Please give some reasoning and explanations in this paper, and add some comparative analysis of different methods.

(3)  In the conclusion section, it is recommended that the authors add some prospects for future research.

(4)  In the article, there are many language and grammatical errors, and then please is polish language and adjust content.

(5)  In this section of Introduction, some more important references on grading nursing care evaluation should be added, meanwhile add and sort out literatures on grading nursing care and some methods, and then give some targeted reviews.

Author Response

Thank you very much for giving us an opportunity to revise our manuscript. We appreciate the editor and reviewers very much for their constructive comments and suggestions on our manuscript entitled“Grading Nursing Care Study in Integrated Medical and Nursing Care Institution Based on Two Stage Grey Synthetic Clustering Model under Social Network Context”.The main corrections in the paper and the responds to the reviewer’s comments are as flowing:

  1. What is the rationale for combining the these approaches of interval-valued intuitionistic fuzzy entropy, two-stage grey synthetic clustering model, and social network? please give some explanations in this paper.

Response:Thank you for your suggestion. To address you concerns, related description has been added in the third section, as follows:

The social network is used to determine the weights of decision makers, and the interval-valued intuitionistic fuzzy method is used to determine the weights of decision indicators. Finally, two stage grey synthetic clustering grading evaluation model is used to determine the care level of the elderly. Social network provides a good solution for the determination of expert weight in interval intuitionistic fuzzy method. The combination of the above methods reduces the uncertainty of index weight in grey model to some extent, Therefore, the combination of the above three methods has good applicability for the application scenario of this study. The specific steps are shown in Section 2.

  1. What is the advantages, validity and scope of the model constructed? Please give some reasoning and explanations in this paper, and add some comparative analysis of different methods.

Response:Thank you for your comments. To address your concerns, we have added the applicability and advantages of the method in section 2.2, and it has been marked in red.

In the process of obtaining data, when judging the physical condition of the elderly, due to the subjectivity of medical staff and the instability of the elderly's health condition, the observed data have strong complexity and uncertainty, and it is often impossible to get specific values, which is called grey.Therefore, the grey theory method is adopted.

  1. In the conclusion section, it is recommended that the authors add some prospects for future research.

Response:Thanks for your comments. To address your concerns, prospects for future research has been added in the conclusion section.

This study provides a new way of thinking for the ability evaluation of the elderly in medical and nursing institutions, which has certain practical significance and theoretical value. At present, medical and nursing integration is in a high-speed development stage, and more mature contents will emerge in the future. Therefore, further research in the future will focus on the following aspects:

  • Customer needs. When obtaining customer needs, we adopt the questionnaire method, which is subjective and difficult to obtain samples. Therefore, it is necessary to combine big data technology, regional differences, gender differences, economic differences, etc., to establish a multi-dimensional customer needs analysis method that conforms to the characteristics of medical and nursing services.
  • Service resource planning. At present, with the limited resources, this study assesses the elderly with different abilities, and can study the planning of medical care and pension resources according to their needs.

  1. In the article, there are many language and grammatical errors, and then please is polish language and adjust content.

Response:Thanks for your comments. To address your concerns, we have thoroughly checked through the writing of the whole manuscript, and tried our best to correct grammatical errors and polish the language. For example, “The difference between integrated medical and nursing care institutions and ordinary nursing care institutions is that the health status of most elderly people is relatively low, so it is more important to pay attention to the rational allocation of medical resources.” has been revised to “Differences between integrated medical and nursing care institutions and ordinary nursing care institutions lie in that the health condition of most elderly people is relatively poor, so it is significant to pay attention to rational allocation of medical resources.” And “Karakoç (2019) used grey cluster analysis to evaluation the development level of provinces;” has been revised to “Karakoç (2019) used grey cluster analysis to evaluate the development level of provinces”.

  1. In this section of Introduction, some more important references on grading nursing care evaluation should be added, meanwhile add and sort out literatures on grading nursing care and some methods, and then give some targeted reviews.

Response:The authors have searched and added the existing research on nursing grading evaluation in the introduction section, which has been marked red.

For the nursing care institutions, there are also many researches that deeply discuss the grading evaluation. Li X Y et al (2019) established a grading indicator system for long-term care of the elderly in medical and nursing care institutions in four aspects: basic action ability, physical function, mental and psychological state, and social function. Li W T (2020) used cluster analysis and AHP method to draw the conclusion that it is most reasonable to divide the elderly in nursing care institutions into four care levels. Shi (2018) established a synthetic classification evaluation model of long-term care for the elderly from seven aspects, such as daily living ability, cognitive ability, medical care, and pressure ulcer risk. Some local governments have also established a grading nursing care evaluation system for nursing care institutions. In 2018, Shanghai put forward the “Shanghai Unified Needs Assessment and Service Management Measures for Elderly Care”, which divided the elderly into six nursing grades based on the dimensions of self-care ability and disease severity. Researchers have conducted in-depth discussions on various aspects of grading nursing evaluation, but there are still some shortcomings. Firstly, studies have evaluated medical and elderly services in a synthetic way, which is not conducive to the separate allocation of resources for medical and nursing care services, and can lead to unreasonable resource allocation. Secondly, most of the studies focused on the construction of indicator systems, ignoring the importance of indicator weights for the grading nursing care. Finally, due to the complexity and uncertainty of the elderly's physical condition, the problem of poor information and uncertainty is widespread, so there is a great limitation in taking the exact figures to evaluate.

Round 2

Reviewer 1 Report

The authors addressed my concern. Its similarity index is up to 29%, nevertheless. I recommend the author revise this document again to lower the similarity index.

Good luck! 

Author Response

We would like to express our appreciation to the Editor and reviewers for carefully reviewing the paper and putting forward constructive suggestions for improving its quality and presentation. In this round, we have tried our best to lower the similarity index. Thanks again for your suggestions.

Reviewer 2 Report

The revised paper has been revised and adjusted accordingly according to the comments and suggestions, and then I am quite satisfied with the revised paper.

Author Response

We would like to express our appreciation to the Editor and  reviewers for carefully reviewing the paper and putting forward constructive suggestions for improving its quality and presentation. Thanks again for your appreciation.